



# Multi-grid algorithm for passive tracer transport in NEMO ocean circulation model: a case study with NEMO OGCM (version 3.6)

Clément Bricaud[1], Julien Le Sommer[2], Gurvan Madec[3], Christophe Calone[4], Julie Deshayes[5], Christian Ethe[6], Jérôme Chanut[7], and Marina Levy[8]

[1]Mercator Ocean International, F-31520 Ramonville-Saint-Agne, France
[2]Univ. Grenoble Alpes, CNRS, IRD, G-INP, IGE, F-38000 Grenoble, France
[3]Laboratoire d'Océanographie et du Climat: Expérimentations et Approches Numériques (LOCEAN), IPSL, Sorbonne Université, Paris, 75005, France
[4]Univ. Grenoble Alpes, CNRS, IRD, G-INP, IGE, F-38000 Grenoble, France
[5]Laboratoire d'Océanographie et du Climat: Expérimentations et Approches Numériques (LOCEAN), IPSL, Sorbonne Université, Paris, 75005, France
[6]Laboratoire d'Océanographie et du Climat: Expérimentations et Approches Numériques (LOCEAN), IPSL, Sorbonne Université, Paris, 75005, France
[7]Mercator Ocean International, F-31520 Ramonville-Saint-Agne, France
[8]Laboratoire d'Océanographie et du Climat: Expérimentations et Approches Numériques (LOCEAN), IPSL, Sorbonne Université, Paris, 75005, France

**Correspondence:** Bricaud Clément (cbricaud@mercator-ocean.fr)

**Abstract.** Ocean biogeochemical models are key tools for both scientific and operational applications. Nevertheless the cost of running these models is often expensive because of the large number of biogeochemical tracers. This has motivated the development of multi-grid approaches where ocean dynamics and tracer transport are computed on grids of different spatial resolution. However, existing multi-grid approaches to tracer transport in ocean modelling do not allow to compute ocean dynamics and tracer transport simultaneously. This paper describes a new multi-grid approach developed for accelerating the computation of passive tracer transport in the NEMO ocean circulation model. In practice, passive tracer transport is computed at runtime on a grid with coarser spatial resolution than the hydrodynamics, which allows to reduce the CPU cost of computing the evolution of tracer. We describe the multi-grid algorithm, its practical implementation in the NEMO ocean model, and discuss its performance on the basis of a series of sensitivity experiments with global ocean model configurations. Our experiments confirm that the spatial resolution of hydrodynamical fields can be coarsened by a factor 3 in both horizontal directions without significantly affecting the resolved passive tracer fields. Overall, the proposed algorithm yields a reduction by a factor 7 of the overhead associated with running a full biogeochemical model like PISCES (with 24 passive tracers). Propositions for reducing further this cost without affecting the resolved solution are discussed.

*Copyright statement.* TEXT



# 1  Introduction

Ocean biogeochemical and ecological models are key tools for numerous applications in oceanography . They are in particular used for process studies (Resplandy et al. (2019)), for climate projections (Rickard et al. (2016), Cabré et al. (2015)), for operational forecasts (Ford et al. (2018)) and for monitoring of marine ecosystems (Fennel et al. (2019)). In practice, ocean biogeochemical and ecological models describe the evolution of the concentration of several tracers which are both transported by oceanic flows and interacting through non-linear relations among each other (Gruber and Doney (2019), Doney and Glover (2019)). The influence of advection on tracer concentration is represented explicitly using velocity fields from ocean circulation models (Chassignet et al. (2019)), which may be run simultaneously with the ocean biogeochemical and ecological models (coupled models).

The computational cost of running biogeochemical and ecological models is usually significant because of the large numbers of variables (Séférian et al. (2016)) required for describing biogeochemical cycles and ecological diversity. This computational cost is split between the computation of tracer transport and the computation of the nonlinear functions relating the state variables of the biogeochemical or ecological models. In coupled applications, the extra-computational cost (aka overhead) with respect to running ocean circulation models only is such that coupled applications are rarely run at the finest grid resolution accessible to ocean circulation models. Typical resolutions of global hydrodynamical models are between $1/4°$ and $1/20°$ whereas typical resolutions of global biogeochemical models are between $1°$ and $1/4°$.

Therefore accounting for the widest possible range of scales in biogeochemical and ecological models is becoming a key bottleneck in the design of operationnal and climate models. Indeed, the role of ocean fine scale (1-200km) dynamics and scale interactions on ocean biogeochemistry and marine ecosystems is now well documented (Lévy et al. (2018)). Ocean fine scale dynamics is known to affect the structure of ecosystem (d'Ovidio et al. (2015)) and to impact the response of ocean biogeochemical cycles to environmental changes (Dufour et al. (2013)). These findings are motivating the ongoing increase in the spatial resolution of ocean components of climate and operational models (Hewitt et al. (2017)). The complexity of biogeochemical and ecological models is also steadily increasing in order to better account for biogeochemical cycles and ecological diversity (Ward et al. (2012)). In this context, improving the computational performance of biogeochemical or ecological models is becoming an important challenge.

Several approaches, including multi-grid methods, have been proposed in the literature for accelerating the integration of biogeochemical and ecological models. Methods have for instance been proposed for accelerating the spin up of complex biogeochemical models. This includes in particular the transport matrix method (Kvale et al. (2017)). A *multi-grid approach* has also been proposed by (Aumont et al. (1998)), where the output from the ocean circulation model used to drive the biogeochemical model are coarse-grained, so that the biogeochemical model runs at a lower spatial resolution. This method is currently implemented for NEMO ocean model output and used in Copernicus global biogeochemical forecasting system (Perruche et al. (2016)).

Interestingly, the notion of effective resolution of physical ocean model solutions provides a theoretical justification for the performance of multi-grid methods for tracer transport. Indeed, because of numerical dissipation and sub-grid closures,



resolved flows properties usually underestimate energy and variance with respect to observations at scales smaller than 5 to
10 times the grid spacing. This observations is formalized with the notion of *effective resolution* introduced by (Skamarock
(2004)). The effective resolution can been seen as the smallest scale that is not affected by numerical dissipation (Soufflet et al.
(2016)). In contrast, numerical resolution refers to the typical resolution of the discrete grid used for numerical integration.
In a series of sensitivity experiment,(Lévy et al. (2012)) shows very little differences in simulated tracer distribution if tracer
transport is computed with velocity field where scale inferior to 5-10 dx have been filtered.

Because the existing implementation of multi-grid tracer transport for NEMO ocean model is not meant to be run at the
runtime with the physical ocean component, its application is strongly limited by the I/O requirements and storage capacity.
Indeed, the method proposed by Aumont et al. (1998) is not a priory designed for *coupled* ocean biogeochemical models
but rather restricted to *offline* applications where the biogeochemical model is using output from a previous ocean circulation
model integration. With this algorithm, data from the ocean circulation model should be output and stored on a local file system.
The frequency of the coupling between ocean dynamics and the biogeochemical model is therefore limited by the frequency
of available ocean model output. The I/O and storage capacity required for such applications is a strong limiting factor with
existing climate and operational systems. It is all the more problematic with the foreseen increase in spatial resolution of ocean
components of climate and operational systems.

This paper describes a new multi-grid algorithm for tracer transport, its implementation in NEMO v3.6 and documents its
performance in a series of global ocean model experiments. We describe in particular in detail what specific choices have
been made for the representation of vertical diffusive fluxes. The algorithm is then tested in a series of global ocean model
simulations of varying complexity.

## 2   Evolution equation for a passive tracer

The objective of the multi-grid algorithm is to solve the evolution equation for a passive tracer on a grid which horizontal spatial
resolution is coarser than that of the associated hydrodynamical model. The passive tracer grid dimensions and hydrodynamic
variables used in this equation need to be computed from the dynamic grid. In this section, we present the evolution equation
for a passive tracer in order to identify which terms are provided by hydrodynamical model and which terms are computed by
the transport module of the biogeochemical model itself.

Following the formalism of the heat conservation equation from (Madec (2008), equation 2.1.d in section 2.1.1 ), the evolu-
tion of a biogeochemical tracer can be written as:

$$\frac{\partial C}{\partial t} = -\nabla \cdot (C \overrightarrow{V}) + D + F + Sink + Sources \tag{1}$$

where $\frac{\partial C}{\partial t}$ represents the tracer tendency, $-\nabla \cdot (C\overrightarrow{V})$ the transport of the tracer C by the hydrodynamic flow, $D$ the
parametrizations for sub-grid-scale processes, $F$ the surface forcing terms (air-sea or lateral boundary fluxes of tracer, if any)
and $Sink$ and $Sources$ which are provided by the biogeochemical model.





The influence of sub-grid-scale processes $D$ can be decomposed in a horizontal part $D^l$ and a vertical part $D^v$:

$$D^l = \nabla_{i,j} \cdot (A^l R \nabla C) \tag{2}$$

represents the contribution of the lateral part along geopotential surfaces, $R$ being a tensor of iso-neutral slopes and $A^l$ the lateral eddy diffusivity coefficient , while

$$D^v = \nabla_k \cdot (A^l R \nabla C) + \frac{\partial (A^v \frac{\partial C}{\partial z})}{\partial z} \tag{3}$$

represents the contribution of the lateral part along the vertical direction (first term on the right hand side) and the effect of vertical subgrid physics (second term on the right hand side) with $A^v$ the vertical eddy diffusivity coefficient.

In order to compute Equation 1 in a multi-grid environment, we first need to be able to apply the $\nabla$ operator on the coarsened grid, which requires to define adequately the coarsened grid coordinates and dimensions. As a second step, we have to design an operator to compute the terms which are coming from the dynamical model, such as the ocean velocity $\overrightarrow{V}$, the lateral and

vertical diffusivity coefficient $A^l$ and $A^v$, the iso-neutral slopes $R$ and the surface forcing terms $F$ on the coarsened grid. Other terms of Equation 1 (tracer tendency, sinks and sources) are computed by the biogeochemical model itself, hence provided on the coarsened grid directly.

## 3    Description of the algorithm

The multi-grid algorithm described below is suitable for Arakawa C-type grid and z-level ocean models. Some adjustments

could be performed to adapt it to B-grid ocean models and/or models using different vertical discretizations, but those are not discussed here.

### 3.1    Definition of the coarsened grid

The first step consists in defining the coordinates, the cell dimensions and the land-sea mask of the coarsened grid based on those of the finer grid. The finer grid used in the ocean dynamics and thermodynamics component is called HR hereafter and

the coarser grid in the passive tracer transport component is called HRCRS hereafter. The horizontal coarsening factor used here is 3, a choice that be discussed thereafter (section 6).

As the equations are solved on an Arakawa C-type grid (Figure 1), temperature, salinity, pressure and other akin variables are computed at the cells center $T$ while the velocities are computed at the center of each face of the cell ( $U$ and $V$), normal to the corresponding velocity component.



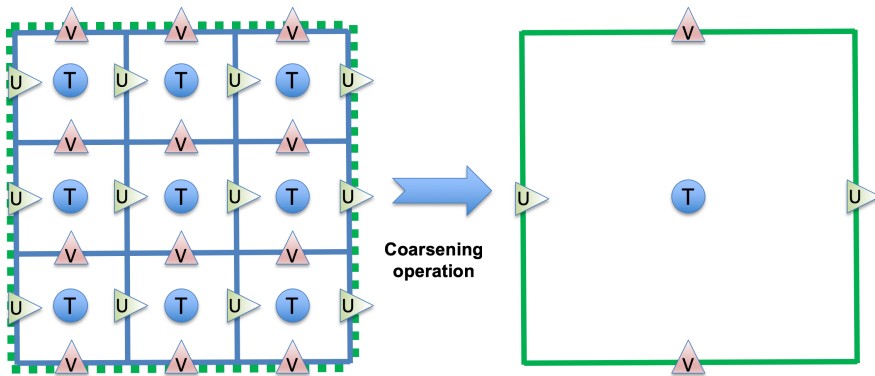

**Figure 1.** Schematic diagram illustrating the definition of the coarsened grid for a coarsening factor of 3 (ocean high-resolution grid is to the left, coarsened grid for biogeochemical model to the right) and localization of tracer (T, blue dots), zonal velocity (U, green triangles) and meridional velocity (V, red triangles) points in both grids. Grid sections that are common to both grids are colored in green.

### 3.1.1 Coordinates

In order to build an HRCRS cell, we take a $3 \times 3$ square of HR cells (Figure 1) and colocate the HRCRS T,U and V grid points at the same location as the corresponding HR grid points. As a result, the definition of HRCRS grid coordinates can be seen as a sub-sampling of HR grid coordinates.

### 3.1.2 Horizontal dimensions

Defining the horizontal dimensions of the coarsened grid cells is necessary to compute gradient or divergence operators.

For a coarsening factor 3, HRCRS zonal cell size is computed by summing 3 consecutive HR zonal cell sizes along the zonal direction, and HRCRS meridional cell size is computed by summing 3 consecutive HR meridional cell sizes along the meridional direction (Figure 2).

### 3.1.3 Land-sea mask

Defining the land-sea mask for HRCRS is the most strenuous task of the multi-grid algorithm, which requires ad-hoc verification as it can lead to substantial errors in the final outputs. If there is at least one ocean T-point in the HR $3 \times 3$ pad, then the corresponding HRCRS T-point will be considered as ocean, which can be defined automatically (Figure 3, top pannels). At the interface between two coarsened grid cells,we carefully retain the shape of the HR mask by adjusting the mask at U- or V-points: when there is at least one ocean point in the HR interface (as in Figure 3, middle pannels), then the interface of the HRCRS corresponding cells is defined as in the ocean. On the contrary, even if the two HRCRS T-points are in the ocean, the interface between them may be defined as on the land, in case the corresponding HR points along the interface are all on land (Figure 3, bottom pannels).



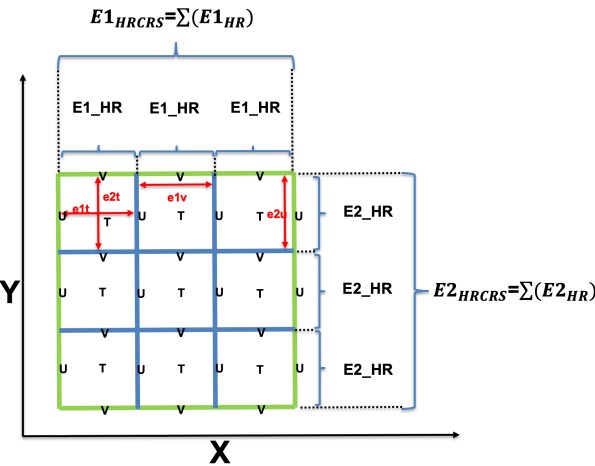

**Figure 2.** Horizontal dimensions of high-resolution ocean HR grid cells (blue lines, red arrows, with names following NEMO nomenclature) and the corresponding coarsened biogeochemical HRCRS grid cell (green lines).

### 3.1.4 Vertical dimensions

The vertical dimension of cells at T-points (usually named $e3t$ in NEMO) are used in vertical divergence and laplacian operators throughout the biogeochemical code. For the divergence operator, it is important that the coarsening procedure preserves the HR grid volume. Hence the vertical dimension of HRCRS grid cells $e3t_{HRCRS}$ is defined as the sum of the volume of the corresponding HR grid cells divided by the HRCRS grid cell horizontal surface:

$$e3t_{HRCRS} = \frac{\sum\limits_{i,j \in HR} e1t * e2t * e3t}{\sum\limits_{i,j \in HR} e1t * e2t} \qquad (4)$$

as illustrated in Figure 4 (top pannels). Note that we assume that HR grid points which are land-masked have their vertical dimension equal to $0$.

For the vertical gradient operator, the coarsening procedure must preserve the HR grid thickness. Therefore, we define another vertical dimension for HRCRS grid cells, $e3tmax$, as the maximum of the vertical dimensions of the corresponding HR grid cells:

$$e3tmax_{HRCRS} = \max_{i,j \in HR} e3t \qquad (5)$$

as illustrated in autorefFigure4 (bottom pannels).



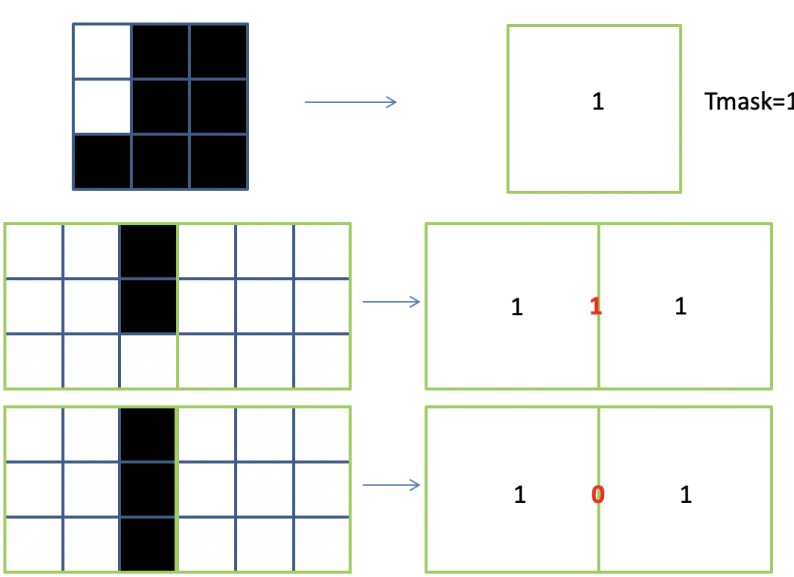

**Figure 3.** Schematic diagram illustrating the definition of land-sea mask from high resolution ocean grid cells (to the left, blue lines with land cells filled in black) to the corresponding coarsened biogeochemical grid cells (to the right, green lines, with numbers identifying whether the corresponding point, here T or U, is defined as land (0) or ocean (1)).

### 3.2 Definition of coarsening operators

Once the coarser grid variables have been defined, the second step of multi-grid algorithm consists in defining the operators used to coarsen the dynamic fields necessary to solve the passive transport equation.

#### 3.2.1 Coarsening of state variables

140 In order to preserve conservation properties through coarsening operations, the intensive state variables (temperature, salinity) of the hydrodynamic model are coarsened such as their volume integrated quantities is the same on the finer and coarser grid. Therefore, the operator is a weighted mean over the $3 \times 3$ pad of HR cells corresponding to each HRCRS cell, where the weights are defined as the volume of the HR grid cells:

$$X_{HRCRS} = \frac{\sum\limits_{i,j} e1t_{HR} * e2t_{HR} * e3t_{HR} * X_{HR}}{\sum\limits_{i,j} e1t_{HR} * e2t_{HR} * e3t_{HR}} \tag{6}$$





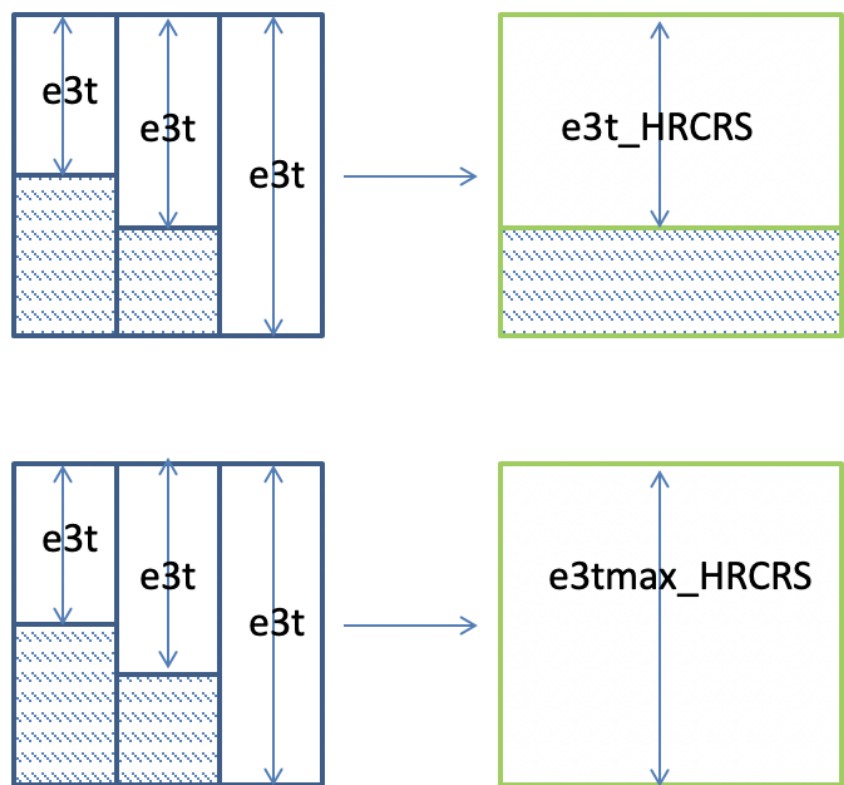

**Figure 4.** Schematic diagram illustrating the coarsening procedure for the vertical dimension (e3t) of grid cells, from high resolution ocean grid cells (blue lines, left pannels) to the corresponding coarsened biogeochemical grid cells (green lines, right pannels).

145    In order to preserve the non-divergence of velocities on the coarser grid, HRCRS velocities are computed so as to conserve horizontal fluxes at the edges of the cells. Hence, the horizontal velocities are coarsened using a weighted mean along the edges of the $3 \times 3$ pad of HR cells corresponding to each HRCRS cell, where the weights are defined as the area of the vertical faces of the HR grid cells:

$$X_{HRCRS} = \frac{\sum\limits_{l} e12_{HR} * e3x_{HR} * X_{HR}}{\sum\limits_{l} e12_{HR} * e3x_{HR}} \qquad (7)$$

150    where $e12_{HR} = e2u_{HR}$, $e3x_{HR} = e3u_{HR}$ and $l = j$ for zonal velocities, and $e12_{HR} = e1v_{HR}$, $e3x_{HR} = e3v_{HR}$ and $l = i$ for meridional velocities. Note that it is the effective horizontal velocities that are coarsened to HRCRS, ie including eventual sub-grid-scale parametrizations, rather than the eulerian velocities.





Vertical velocities and surface fluxes are coarsened using a weighted mean over the $3 \times 3$ pad of HR cells corresponding to each HRCRS cell, where the weights are defined as the horizontal area of HR grid cells:

$$X_{HRCRS} = \frac{\sum\limits_{i,j} e1t_{HR} * e2t_{HR} * X_{HR}}{\sum\limits_{i,j} e1t_{HR} * e2t_{HR}} \tag{8}$$

### 3.2.2 Quantities related to the equation of state

Quantities related to the equation of state have a substantial impact on the model stability, via stratification. Therefore, it is necessary to define conscientiously the coarsening procedure of those, so as to prevent the multi-grid algorithm from generating spurious numerical instabilities.

The $N^2$ buoyancy frequency in the HRCRS domain is computed by coarsening the HR buoyancy using the mean operator (6), a solution that is considerably cheaper than computing HRCRS $N^2$ in the HRCRS domain. In parallel, the density in the HRCRS domain is computed from the temperature and salinity fields in the HRCRS domain. Finally, both the buoyancy frequency and density in HRCRS domain are used to compute the HRCRS slopes for neutral surfaces $R$. This ad-hoc solution (note that there is no well defined coarsening operator for a slope) was found to yield less numerical instabilities than others.

### 3.2.3 Sub-grid-scale vertical processes

Vertical processes affecting passive tracers are predominantly led by vertical mixing thus are very sensible to its amplitude. The vertical mixing coefficient $A^v$ has minimal background values of the order of 1.e-5 $m^2.s^{-1}$ or less, but can reach up to 10 $m^2.s^{-1}$ or more at sporadic grid points (when static instability occurs and the parametrization for convection is activated, for example). As a consequence, using a simple weighted mean of the HR $A^v$ as coarsening procedure, is expected to be inappropriate. Indeed, with such method, HRCRS $A^v$ would be exclusively reflecting occurrences of strong values of HR $A^v$, whether they are adjacent to lower ones or not. Henceforth, we define 6 different operators to coarsen the vertical mixing coefficient and test the sensitivity of the multi-grid algorithm to those (see section 5.3.1):

- the MIN and MAX operators, where the coarsened $A^v$ is the minimum or, respectively, the maximum of the HR $A^v$ over the $3 \times 3$ pad of HR cells corresponding to each HRCRS cell;

- the MEAN operator, where the coarsened $A^v$ is computed by a weighted mean in a $3 \times 3$ pad of HR $A^v$ (ie the method expected to be inappropriate);

- the MEDIAN operator, where the coarsened $A^v$ is the median value in a $3 \times 3$ pad of HR $A^v$;

- the MEANLOG operator, where the coarsened $A^v$ is computed by a weighted mean in logarithmic space in a $3 \times 3$ pad of HR $A^v$.





### 3.3 Practical implementation in NEMO ocean model

By default (i.e without using multi-grid algorithm), the model passive transport component uses the grid and dynamic variables computed by the dynamic component of the code (i.e on the HR grid). It is necessary to implement the multi-grid algorithm for passive tracer transport in the NEMO OGCM without cancelling the default option. We describe here how the model passive transport component can switch to the HRCRS grid and dynamic variables when the multi-grid algorithm is activated.

First we present how we manage the domain loop indices. The loops in the ocean dynamic and passive transport components uses the domain sizes in the three spatial directions (called $(jpi, jpj, jpk)$ in NEMO). When the multi-grid algorithm is activated, HR grid sizes $(jpi, jpj, jpk)$ are saved in other variables $(jpi_{HR}, jpj_{HR}, jpk_{/}HR)$ and HRCRS coarsened grid sizes are defined as $(jpi_{HRCRS}, jpj_{HRCRS}, jpk_{HRCRS})$. When the code is in the ocean dynamic component, the loop sizes $(jpi, jpj, jpk)$ are set to HR grid sizes $(jpi_{HR}, jpj_{HR}, jpk_{HR})$. When the code enters the passive transport component, the loop sizes $(jpi, jpj, jpk)$ switch to HRCRS grid sizes $(jpi_{HRCRS}, jpj_{HRCRS}, jpk_{HRCRS})$ ; when the code goes out of the passive transport component, the loops sizes $(jpi, jpj, jpk)$ switch back to HR grid sizes:

---

**Algorithm 1** Algorithm to switch domain loop indices from finer to coarser grid

---

**Initialization**

...

Define HR grid sizes : $(jpi_{HR}, jpj_{HR}, jpk_{HR})$

Define HRCRS grid sizes : $(jpi_{HRCRS}, jpj_{HRCRS}, jpk_{HRCRS})$

Define loop indices for whole code : $(jpi, jpj, jpk)$

...

**Step**

...

$(jpi, jpj, jpk) = (jpi_{HR}, jpj_{HR}, jpk_{HR})$

**CALL DYNAMIC component**

...

Switch to HRCRS : $(jpi, jpj, jpk) = (jpi_{HRCRS}, jpj_{HRCRS}, jpk_{HRCRS})$

**CALL PASSIVE TRACER component**

Switch to HR : $(jpi, jpj, jpk) = (jpi_{HR}, jpj_{HR}, jpk_{HR})$

...

---

Secondly, we present how we manage the grid and dynamic variables used in the passive transport component. They are called to memory through pointers defined in an interface between the dynamical and passive transport components (called $oce\_trc$ in NEMO GCM):

To avoid the duplication of coarsened and non coarsened grid and dynamic variables in the passive transport routine, the pointers target the coarsened grid and dynamic variables when the multi-grid algorithm is activated and the non coarsened





---

**Algorithm 2** Algorithm to link velocities in passive tracer transport component

---

**MODULE passive tracer**

USE *oce_trc*, ONLY: U,V,W !interface between dynamic and passive transport components

**CONTAINS**

**SUBROUTINE** *operator_passive_tracer*

...

$X = operator(U, V, W)$

...

**END SUBROUTINE** *operator_passive_tracer*

**END MODULE** *passive_tracer*

---

variables when it is not activated. The choice between the two definitions is done at the pre-compilation stage of the code with a cpp key $key\_crs$:

---

**Algorithm 3** Algorithm to avoid duplication in passive transport routines

---

**if defined** $key\_crs$ ! multi algorithm activated:

target are coarsened (HRCRS) velocities

declared in coarsened variables module:

USE crs, ONLY: $U => U_{HRCRS}, V => V_{HRCRS}, W => W_{HRCRS}$

**else** ! multi-grid algorithm not activated:

target are non coarsened (HR) velocities

declared in oce module:

USE oce, ONLY: $U => U, V => V, W => W$

**endif**

---

## 4 Experimental protocol

The multi-grid algorithm has been implemented in NEMO GCM (Madec (2008)) version 3.6 in which the OPA ocean model is coupled to LIM3 sea ice model (Rousset et al. (2015),Vancoppenolle et al. (2009)) and its passive tracer transport component TOP.





Three different global ocean configurations have been employed to assess the multi-grid algorithm:

- the HR configuration, where the ocean hydrodynamic (ie open ocean and sea ice) and the passive tracer components are both running at $1/4°$ resolution. This is the reference experiment using the highest possible resolution and no multi-grid algorithm for passive tracers.

- the LR configuration, where the ocean hydrodynamic and the passive tracer components are both running at $3/4°$ resolution. This experiment provides insights on the impact of horizontal resolution with still no multi-grid algorithm for
passive tracers.

- the HRCRS configuration, where the ocean hydrodynamic component is running at $1/4°$ resolution while the passive tracer is running at $3/4°$ resolution. This experiment corresponds to the standard use of multi-grid algorithm for passive tracers.

It is important to state upfront that it is not expected that the HRCRS configuration yields exactly the same results as the HR
nor LR configurations. Indeed, as there is some information missing in the coarsening procedure of ocean hydrodynamics from HR to HRCRS, the passive tracer distribution in HRCRS is not expected to be strictly the same as that in HR configuration. On the other hand, because the hydrodynamics in LR is running at coarser resolution than that in HRCRS, it is not expected that the passive tracer distribution in HRCRS is the same as that in LR. Yet, the multi-grid algorithm will be proven useful if the results of HRCRS are closer to HR than those of LR. Hence the present experimental protocol should not be seen as a direct
validation of the multi-grid algorithm, but rather as an assessment of its potential benefits and limitations.

### 4.1 Model configurations

The $1/4°$ configuration is described in Barnier et al. (2006). The $3/4°$ configuration coordinates and bathymetry are built by applying on $1/4°$ configuration coordinates and bathymetry the coarsening operators described above. This operation has created some fake straits in Panama and the Indonesian Trough Flow, which have been manually filled. The $3/4°$ configuration
is closed to the global Ocean ORCA $1°$ configuration used in various CMIP5 (Voldoire et al. (2013)) and CMIP6 models (Voldoire et al. (2019)).

The $1/4°$ and $3/4°$ resolution configurations are based on a quasi isotropic ORCA grid (Madec and Imbard (1996)) at nominal $1/4°$ and $3/4°$ horizontal resolution, respectively, at the equator. Vertical discretization uses 75 z-levels with partial cell parametrization (Barnier et al. (2006)), which leads to a resolution ranging from 1 meter at the surface to 450 meters at
depth.

The bathymetry is based on a combination of ETOPO1 (Amante and Eakins (2009)) and GEBCO08 (Becker et al. (2009)). The initial state for the ocean is at rest, with temperature and salinity from WOA13 (World Ocean Atlas / www.nodc.noaa.gov). At the surface, the model is forced by ERAinterim atmospheric reanalysis (Dee et al. (2011)) using the CORE bulk formulae (Large and Yeager (2004)) to compute the turbulent air-sea fluxes. A monthly runoff climatology is built and applied to all
configurations, using data for coastal runoffs and major rivers from Dai and Trenberth (Dai and Trenberth (2002)).





| resolution | viscosity: $B_o^l$(m$^4$/s) | diffusion: $A_o^l$(m$^2$/s) |
|---|---|---|
| $1/4°$ | -3.2e11 | 300. |
| $3/4°$ | -86.4e11 | 900. |

**Table 1.** Horizontal eddy viscosity and diffusivity values at the Equator.

A split-explicit time-splitting scheme is employed to compute the surface pressure gradient with a non-linear free surface (Levier et al. (2007)). These parametrizations were already used in the Irish-Biscay-Iberian regional configuration, described in Maraldi et al. (2013).

The momentum advection term is computed with the energy and enstrophy conserving scheme proposed by Arakawa and
Lamb (Arakawa and Lamb (1981)). The advection of tracers (temperature and salinity) is computed with a total variance diminishing (TVD) advection scheme (Lévy et al. (2001), Cravatte et al. (2007)).

Lateral eddy viscosity is computed with a biharmonic operator and lateral eddy diffusivity is computed with an isopycnal laplacian operator for both active and passive tracers. As explained in Madec (2008) (section 9.1), lateral eddy diffusivity coefficient $A^l$ decreases proportionally to the grid size, while the lateral eddy viscosity coefficient $B^l$ decreases poleward as
the cube of the grid cell size:

$$A^l = \frac{\max(e1,e2)}{e_{max}} A_o^l \tag{9}$$

$$B^l = \frac{\max(e1,e2)^3}{e_{max}^3} B_o^l \tag{10}$$

where $A_o^l$ and $B_o^l$ are the respective values at the Equator (see table Table 1) while $e_{max}$ is the maximum of e1 and e2 taken
over the whole masked ocean domain. The lateral eddy diffusivity coefficients shown here are used for active tracers when the grid multi-grid algorithm is activated or not, and used for passive tracers when the grid multi-grid algorithm is not activated; the choice of lateral eddy diffusivity for passive tracers when the multi-grid algorithm is activated, is discussed later (section subsection 5.2).

The vertical eddy viscosity and diffusivity coefficients are computed from a TKE turbulent closure model (Blanke and
Delecluse (1993)), with parameters as described in Reffray et al. (2015).

### 4.2  Numerical experiments

To assess the multi-grid algorithm for passive tracers, three types of experiments have been set up. They differ on the passive tracer initial state or evolution. In all cases, the dynamical component is the same. The duration of all experiments is 365 days.

In the PATCH experiments, the multi-grid algorithm is assessed on the horizontal component of the transport equation (1).
Here, the passive tracer vertical eddy diffusivity component is set to 0 ($A^v = 0.$ ). The passive tracer $C$ is initialized with a





patch of $10°$ radius centered on $60°$West / $36°$North (see top left panel in Figure 6). The value of the tracer ranges from 2 at its center to 1.368 at its boundaries and is vertically uniform. Outside of the patch, the tracer initial value is set to 1. The tracer has no sources nor sinks. Hence the passive tracer evolution in the PATCH experiment can be described by the following equations:

$$
\begin{cases}
\frac{\partial C}{\partial t} = -\nabla \cdot (C\overrightarrow{V}) + \nabla \cdot (A^l R \nabla C) \\
\quad C(t=0) = \text{PATCH}
\end{cases}
\tag{11}
$$

where $R$ is a tensor of iso-neutral slopes.

The AGE-ZDIF experiments are conceived to assess the multi-grid algorithm on the vertical eddy diffusive component of the passive tracer transport equation (1). Hence the advection and lateral diffusion are unplugged for the passive tracer. This experiment makes use of the so-called Age tracer of NEMO GCM: its initial value is set to zero in all the ocean and its value is damped to 0 at the ocean surface while there is a net source in the ocean interior. This tracer gives a good representation

of the time evolution of a water mass, after it was in contact with the Ocean surface ;hence it gives some information on the ventilation of water masses.The tracer evolution in the AGE-ZDIF experiment can be described by the following equations:

$$
\begin{cases}
\frac{\partial C}{\partial t} = \frac{\partial (A^v \frac{\partial C}{\partial z})}{\partial z} + \text{Sources} \\
\quad \text{Sources}\,(t > 0) = 1 \\
\quad\quad C(t=0) = 0. \\
\quad\quad C(z=0) = 0.
\end{cases}
\tag{12}
$$

Finally, the multi-grid algorithm is assessed on the full tracer transport equation (1) in the AGE-FULL experiment which also uses the Age tracer, but retaining all terms for the tracer transport equation. Hence the tracer evolution in the AGE-FULL

experiment can be described by the following equations:

$$
\begin{cases}
\frac{\partial C}{\partial t} = -\overrightarrow{V} \cdot \nabla C + \nabla \cdot (A^l R \nabla C) + \frac{\partial (A^v \frac{\partial C}{\partial z})}{\partial z} + \text{Sources} \\
\quad\quad \text{Sources}\,(t > 0) = 1 \\
\quad\quad\quad C(t=0) = 0. \\
\quad\quad\quad C(z=0) = 0.
\end{cases}
\tag{13}
$$

## 5 Results

### 5.1 Coarsened velocities

Advection by ocean currents is one of the main drivers of the evolution of passive tracers. As a consequence, it is important

that tracer transport by ocean currents on the coarsened grid HRCRS remains close to that on the HR grid. This implies



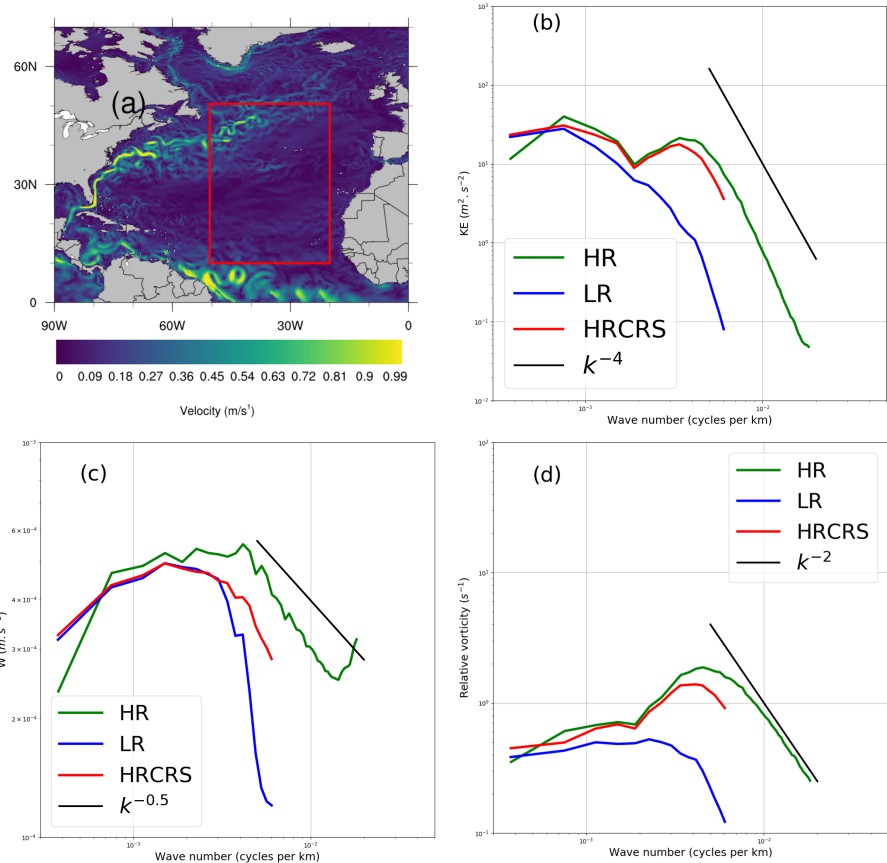

**Figure 5.** North Atlantic yearly mean surface current speed (a) and power spectrum density of (b) surface kinetic energy, (c) vertical velocities and (d) surface vorticity as functions of spatial wave number, in configurations HR (green lines), LR (blue lines) and HRCRS (red lines). Black lines in spectra represent usual slopes to be compared with numerical spectra. All computations use daily model outputs and yearly averaged.

that coarsened velocities share similarities with HR velocities on the spatial scales that are common. To test this, coarsened velocities are compared to HR velocities and LR velocities (Figure 5), as done by Lévy et al. (2012) in a more idealized context.

In terms of kinetic energy, vertical velocities and relative vorticity, HR, HRCRS and LR have almost the same level of energy at spatial scales larger than 250 km. At smaller spatial scales, HRCRS has a level of energy comparable to HR in terms of kinetic energy and relative vorticity while vertical velocities are less energetic than in HR. For the 3 quantities, at spatial scales smaller than 250 km, the level of energy of LR is significantly below HR and HRCRS levels. This suggests that ocean currents in HRCRS are, overall, more similar to those in HR than to those in LR.

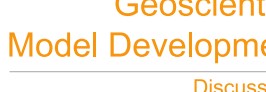

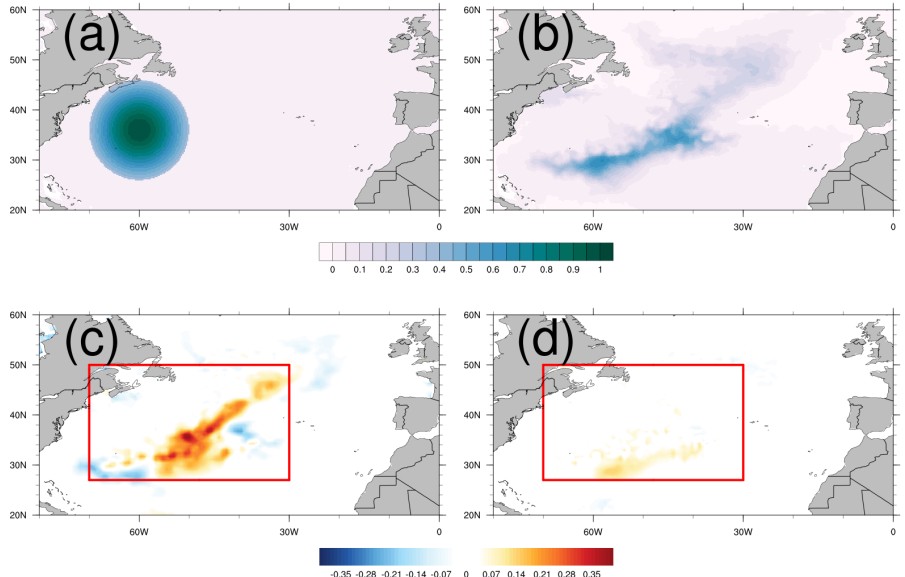

**Figure 6.** Passive tracer Initial condition for PATCH experiment (a), tracer concentration at 16 m depth after one year for HR configuration (b) and difference with respectively LR (c) and HRCRS configurations (d).

## 5.2 Horizontal tracer transport

The evaluation of the multi-grid algorithm to resolve an advection-diffusion problem is shown here with the results of the
PATCH experiment described in section subsection 4.2.

The first step is to evaluate the best value for the passive tracer horizontal diffusion coefficient in the HRCRS configuration. A comparison between HR and various HRCRS configurations differing for their horizontal diffusion coefficient (set to 300, 600, 900 and 1200 $\mathrm{m}^2/\mathrm{s}$) is presented in appendix Appendix A. This sensitivity test suggests that 900 $\mathrm{m}^2/\mathrm{s}$ yields the best results for HRCRS. Using 300 $\mathrm{m}^2/\mathrm{s}$ in HR and 900 $\mathrm{m}^2/\mathrm{s}$ in HRCRS for the passive tracer lateral diffusion respects the ratio in
spatial resolution between the two configurations (of respectively $1/4°$ and $3/4°$ nominal resolution), which is consistent with the linear relationship between the horizontal diffusion coefficient $A^l$ and the cell size (see Equation 9).

Using this value for the lateral diffusion coefficient $A^l$ in the HRCRS configuration, and comparing outputs to those from HR and LR configurations, suggests that the HRCRS patch is closer to the HR shape than to the LR one (Figure 6). Indeed, LR has a larger area where its concentration is higher than in HR, with a difference greater than 0.3 in its center, while the
difference between HRCRS and HR does not exceed 0.2 (within the red box, after one year of simulation, RMSE is 0.04 between HRCRS and HR and 0.09 between LR and HR).

Horizontal resolution plays an important role on the horizontal velocities, that influences the interior EKE distribution and so the tracer distribution. In Figure 7, we represent for a meridional section in the Gulf stream the daily vertical distribution of zonal velocities and PATCH tracer after one year of simulation, for the HR, HRCRS and LR configurations. The HRCRS

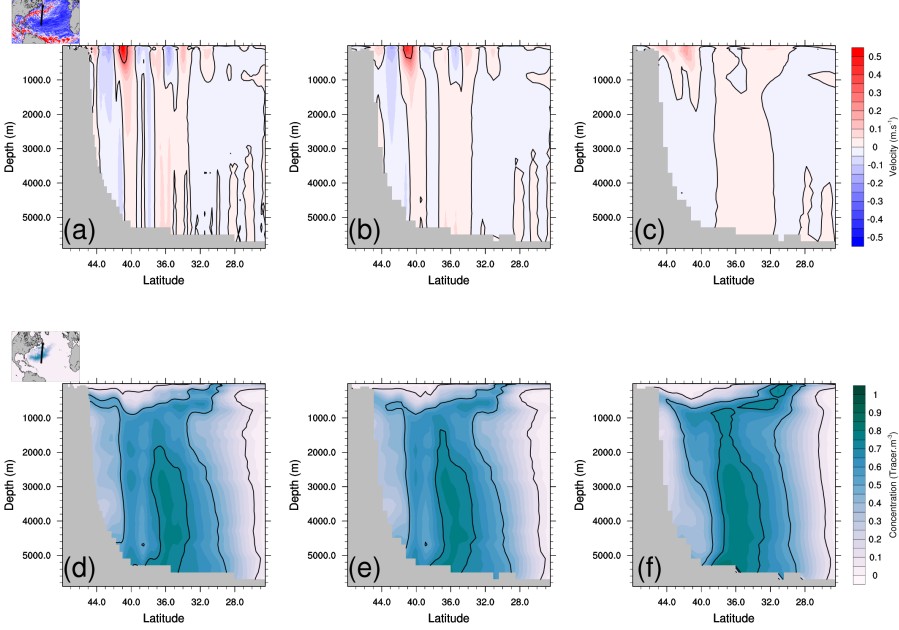

**Figure 7.** Vertical distribution of the zonal velocities ($m/s$) (top) and PATCH tracer (bottom) for HR (a,d), HRCRS (b,e) and LR (c,f) configurations along $55°$W section in Gulf stream.

velocities are obtained by coarsening the HR velocities. HR and HRCRS zonal velocities are stronger than LR velocities and present more fine structures, in particular in the northern region ( above $40°$ North). HRCRS and HR tracer distribution are similar and present some differences to LR configuration. In the southern part of the $55°$W sections, LR configurations has a maximum of concentration, whereas HRCRS and HR have lower values. In the northern parts of the section, HR and HRCRS have some higher values of concentration in the interior and deeper ocean than in LR configuration, around $40°$N latitude. Despite the spatial resolution degradation operated on the velocities, the dynamics used to transport HRCRS tracer present the same pattern as the HR dynamics. The tracer using the multi-grid algorithm reproduces the the HR tracer; while the dynamics is coarsened, the HRCRS tracer benefits from the higher resolution dynamics and present a gain compared to a lower resolution experiment.

## 5.3 Vertical diffusion

### 5.3.1 Choice of vertical diffusion on the coarsened grid

Vertical mixing is responsible of the ventilation of a passive tracers and thus its vertical distribution. The vertical diffusion coefficient $A^v$ is computed by the dynamic component (so on the HR grid) and it might computed on the HRCRS grid.





| configuration | RMSE |
|---|---|
| LR | 17.72 |
| HRCRS with MIN | 14.40 |
| HRCRS with MEANLOG | 10.00 |
| HRCRS with MEAN | 12.03 |
| HRCRS with MEDIAN | 10.46 |
| HRCRS with MAX | 42.16 |

**Table 2.** RMSE with HR solution, after one year of simulation.

Here we present the results of the AGE-ZDF experiment in order to assess the capacity of the multi-grid algorithm to simulate the passive tracer vertical mixing. In particular we show the sensitivity of the Age tracer vertical representation to the choice of coarsening operator of the vertical mixing parameter $A^v$ presented in section section 3.2.4.

The top figures in Figure 8 present the daily vertical profiles of rms difference to the HR Age tracer after one year of simulation. All the configurations have the same shape of vertical distribution, except LR and HRCRS with MAX operator which have a behaviour very different to the others at the bottom of the Ocean. The top right panel in Figure 8 presents a zoom in the first 500 meters. All the configurations have some differences in the mixed layer and some HRCRS configurations are closer to HR, according to the coarsening operator used for the vertical mixing: the HRCRS with MEANLOG and MEDIAN operators give the best performances compared to HR experiment, whereas the HRCRS experiments with the 2 extreme operators gives the worst results with LR.

Table 2 gives the RMSE with HR solution for all HRCRS experiments and LR. The HRCRS with MEANLOG and MEDIAN operators gives the best results. HRCRS with MIN and HRCRS with MEAN have a better skil than LR and HRCRS with MAX is far from the other simulations.

To understand which process leads to these solutions, we present on Figure 8 bottom figures the divergence of the age tracer vertical diffusive fluxes averaged over the global Ocean and the first year of the simulation. A positive divergence means that the net incoming fluxes are positives. Looking at the flux to the bottom (bottom right plot) shows that the divergence is negative for HRCRS with MAX operator; it is coherent with the fact that Age tracer concentration is lower than the other (see top left plot). It is explained by the fact that the MAX operator extents the zone where the $A^v$ has high value along the bathymetry. HRCRS with MIN operator has a negative flux divergence at the bottom, but its Age value is closer to HR.

### 5.3.2 Special case for convection

Here we propose to asses the multi-grid algorithm to simulate the age tracer penetration for deep convection as in Ross and Weddel seas. We want to know if the operators selected in the previous section (MEANLOG and MEDIAN) are appropriate for this particular situation.



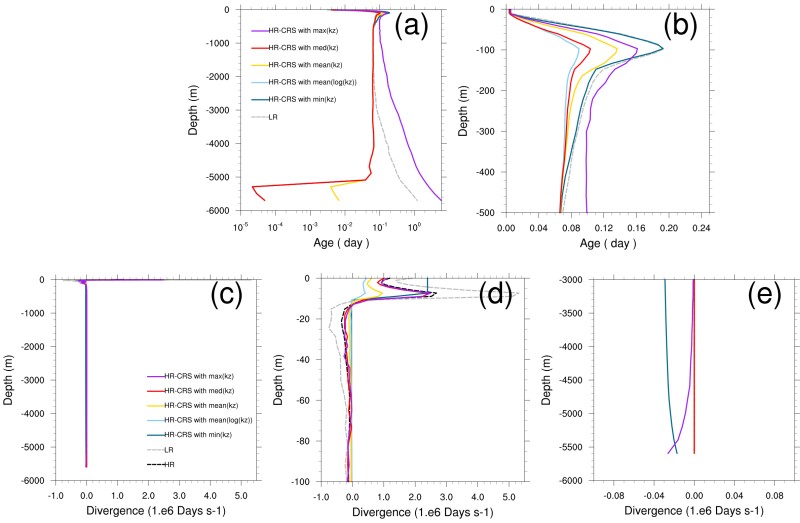

**Figure 8.** Top: RMS differences to HR Age tracer for HRCRS & LR runs for the whole water column (a) and for the first 500 meters (b). Age Tracer values are instantaneous outputs after one years. Bottom: Horizontal mean of divergence of Age tracer vertical diffusive fluxes mean over the first year for the whole water column (c), the top ocean (d) and the bottom ocean (e).

Figure 9 represents the depth where the Age tracer reaches the value of 10 days (top) and 100 days, along a section in the Austral Ocean, at the latitude of $58.3°$ south. On the two plots on the left, we compare the HR profile (black), the LR experiment (grey) and the ensemble spread of the different solutions of HRCRS (light blue). The HR profile is included in the HRCRS ensemble, whereas the LR experiment is outside of the HRCRS spread. All the HRCRS solution have better performances than

LR solution compared to HR. However, the spread is larger in two areas, between $140°$W and $80°$W and between $100°$E and $160°$E.

The 2 plots on the right of Figure 9 shows the differences between each HRCRS solutions and HR. We can see that, outside of the 2 areas of deep convection, the HRCRS solution performed with MEANLOG and MEDIAN operator are the closest to the HR solution, especially in the deeper case ( bottom right figure). Around $100°$E and $150°$W, the HRCRS solution with the

MIN operator is the closest to HR solution.

In order to combine the better performance of MIN operator in convection situation and MEANLOG operator in the global Ocean , we add here a new operator based on MEANLOG operator with a switch to MIN operator in presence of convection. The HRCRS run with MEANLOG operator and a switch to MIN operator gives the best comparison to the HR run in the Weddel and Ross seas areas and have good performances outside.

**5.4    Evaluation of the full algorithm**

We have validated the multi-grid algorithm to represent the tracer advection and lateral diffusion in section $5.2$ and the vertical eddy diffusion in the section $5.3$. The choice of the operator to coarsen the vertical diffusion coefficient $A^v$ has also been



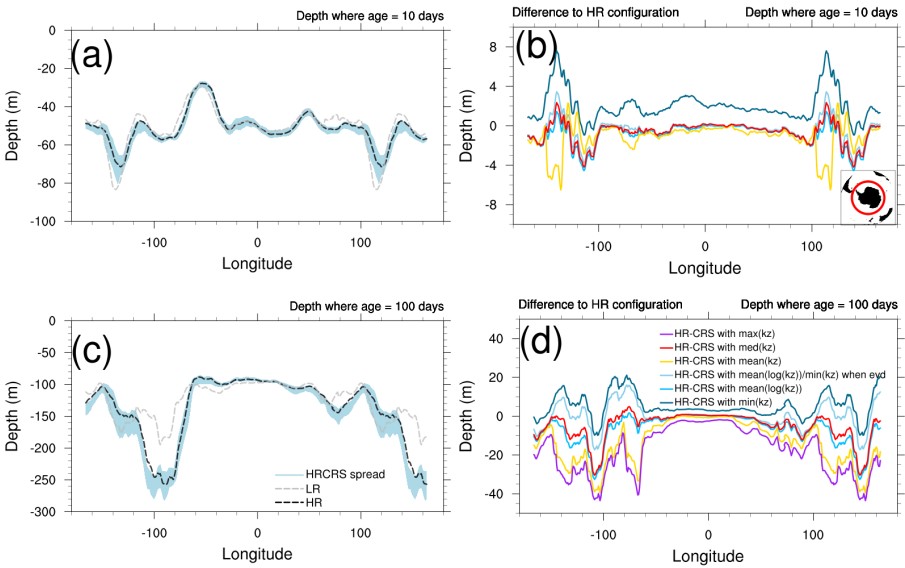

**Figure 9.** ACC section ( $-58.3°$ South ) of depth where age tracer value is equal to a given age: 10 days (a,b) and 100 days (c,d). (a) and (c) compare HR, LR and the mean of HRCRS runs. (b) and (d) show the difference between HR and all the HRCRS runs.

discussed in the section 5.3. Now we present some results for the full transport equation (1). We decide to continue the assessment using the MEANLOG operator to coarsen the vertical diffusion coefficient.

A comparison of the depth for HRCRS and LR runs to HR run is presented in Figure 10. The upper figure presents the HR depths where Age is equal to 100 days. The second figure presents the depth differences between LR and HR and the last figure the depth differences between HRCRS and HR. Higher depths values are due to deeper Age tracer penetration, caused to higher vertical mixing. Globally, there are no big differences between the 3 runs, excepted on the area were mesoscale activity is important: the Gulf Stream, the North Atlantic subpolar gyre, the Kuroshio and the Austral Ocean. The differences between

HRCRS and HR runs are largely weaker than the differences between LR and HR runs.

     In order to compare on depth the three different runs, we present in Figure 11 the Age tracer penetration along 3 sections. The first section starts in ACC and goes northward until North Atlantic. For a value of 10 days, the penetration for the 3 experiments are close. For a value of 100 or 300 days, HRCRS is very close to HR run and LR is different from the 2 others experiments. The second section goes along the Equatorial Pacific. For the 3 values of Age Tracer, the corresponding depth from HR and

HRCRS are very close and the depths from the LR experiment are different. The last section is inside the Antarctic Circumpolar Current. HRCRS and HR depths are similar for 10, 100 and 300 days. LR run depths are similar to the 2 other runs for 10 days, and the differences with the 2 other runs increase with depth.

     The Age tracer RMSE with HR solution has a value of 0.91 for HRCRS and 2.30 for LR, after one year of simulation (Table 3).



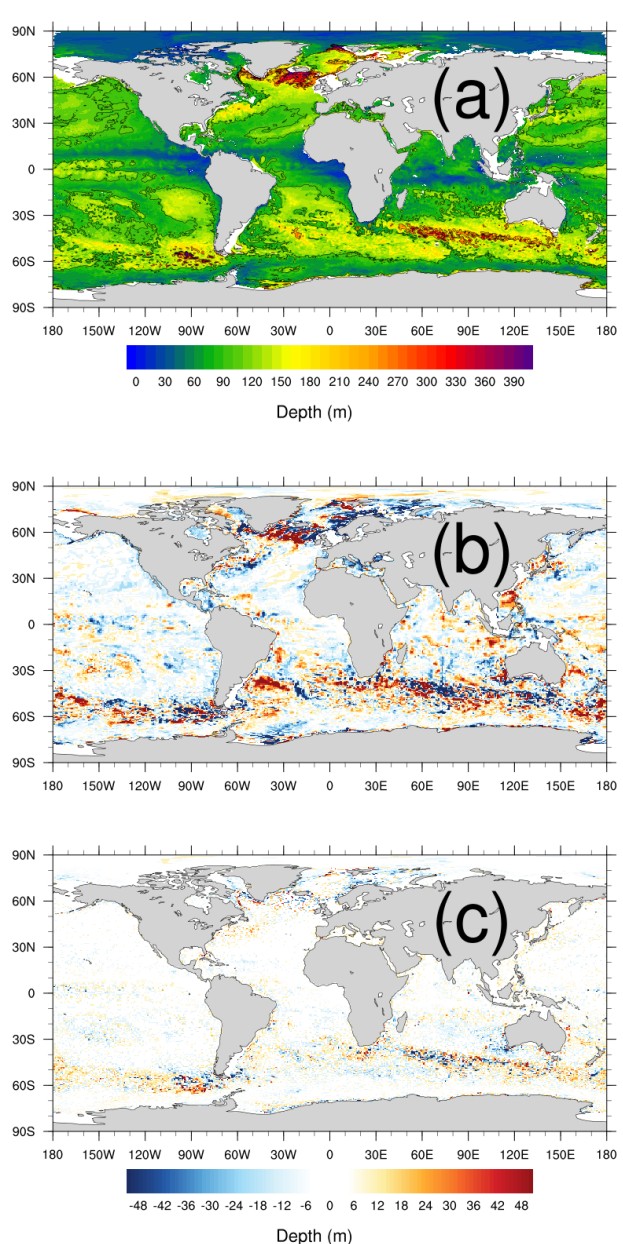

**Figure 10.** Depth where age tracer is equal to 100 days, after one year of simulation: HR experiment (a), LR minus HR (b) and HRCRS minus HR (c).



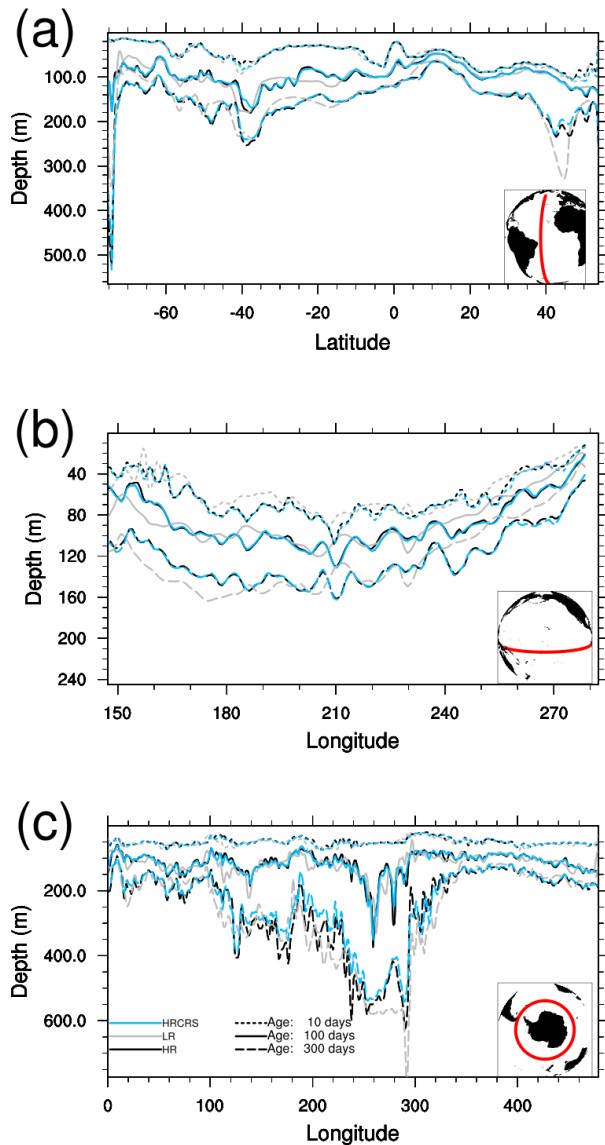

**Figure 11.** Atlantic (a), Pacific (b) and ACC (c) sections of Age tracers after one year of simulation for HR (black), LR (grey) and HRCRS (blue) configurations. Age value is 10 days (dashed line), 100 days (solid line) and 300 days (dashed line).

## 5.5 Computational performance

Because of the extra computational cost of the coarsening of dynamical fields, the multi-grid algorithm procedure gives better performances with large numbers of passive tracer fields. Indeed, the expected speed-up for computing tracer advection on





| configuration | LR | HRCRS |
|---|---|---|
| RMSE | 2.30 | 0.91 |

**Table 3.** RMSE with HR solution, after one year of simulation.

| configuration | HR | HRCRS |
|---|---|---|
| tracer transport module (for 1 tracer) | 168. | 18.7 |
| coarsening module | - | 220. |
| total for 1 tracer | 168. | 238.7 |
| total for 24 tracers | 4032. | 668. |

**Table 4.** Elapsed time for a 1 month run with/without the multi-grid algorithm. The values in the last column have been extrapolated.

a 3x3 coarser resolution grid is a priory a factor 9. However, in practice, the time spent for defining the grid and computing dynamical fields on the coarsened grid should also be accounted for. We present in Table 4 the elapsed time spent by the HR
run (without grid coarsening capacity) and HRCRS run (with grid coarsening capacity). The elapsed time spent defining the scale factors for coarsened grid and computing the dynamic fields on the coarsened grid is actually significant. This is why, for a single tracer field, the HRCRS run (with grid coarsening capacity) uses more CPU resources than the HR run. But it should be noted that the definition of the grid and coarsened dynamical fields is only done once for all the tracer fields. Therefore the net overhead cost of this extra operation is relatively small in situations with many tracer fields (as shown in the last column of
Table 4).

Overall, we estimate that using the multi-grid algorithm allows to reduce the cost of running a full ocean model with an interactive biogeochemical component by a factor ~ 3. Table 5 provides an estimate of the breakdown of elapsed time in a global NEMO configuration coupled to PISCES biogeochemical model (Aumont et al. (2015)), which simulates 24 passive tracers. Typically, running NEMO biogeochemical transport module (TOP) for 24 tracers on the same grid as the dynamics
requires twice as much resources than the ocean/sea-ice component. PISCES biogeochemical component itself uses resources comparable to those used for the ocean/sea-ice component. Running a full model with a biogeochemical component based on PISCES typically requires 4 times the resources of the ocean/sea-ice component alone. Given the figures of table Table 4, we estimate that the net elapsed time overhead of the biogeochemical component (transport + PISCES) would be reduced by a factor ~7 with the multi-grid algorithm. Therefore, a full model using our multi-grid algorithm would use 1.45 times the
resources the ocean/sea-ice component alone, therefore reducing the cost of the overall system by a factor 3.

## 6  Limitations and future perspectives

A key limitation of the multi-grid algorithm described in this paper is its restriction to *odd* coarsening factors. As shown in Figure 12, two possibilities would exist for defining an averaging procedure in the case of *even* coarsening factors but none of




| configuration | HR | HRCRS | speedup ratio |
|---|---|---|---|
| ocean and sea ice | 1. | 1. | 1. |
| coarsening operation | 0. | 0.12 | - |
| BGC tracers transport | 2. | 0.22 | 9. |
| BGC model | 1. | 0.11 | 9. |
| overhead of BGC | 3. | 0.45 | 6.7 |
| total | 4. | 1.45 | 2.8 |

**Table 5.** Typical expected speedup in elapsed time between a run with versus without multi-grid algorithm. Values have been extrapolated from the measured cost of the coarsening operation and an estimate of the net overhead of PISCES biogeochemical model.

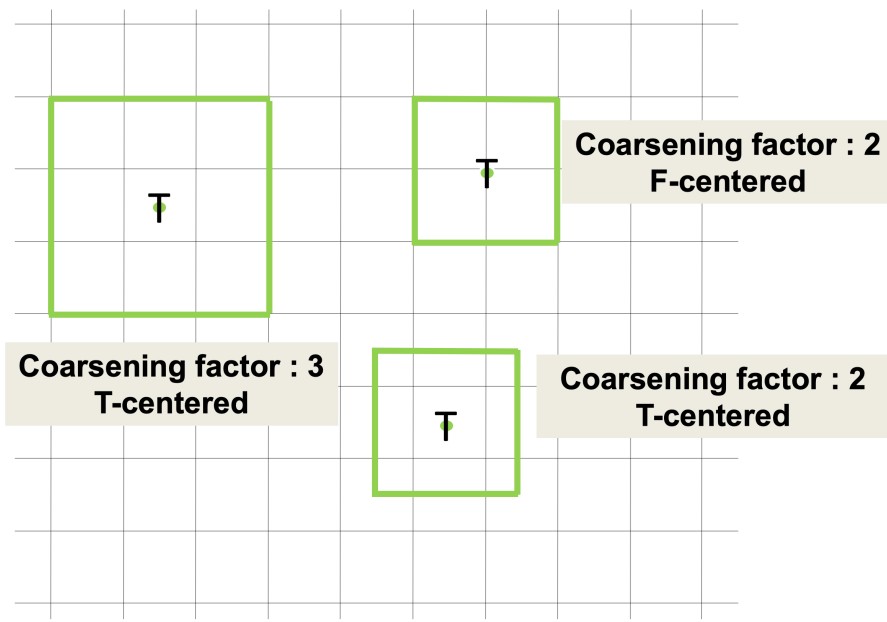

**Figure 12.** Positon of coarsened cells (green) on its high-resolution grid local mpp-domain, for some odd or even coarsening factors.

them would be fully amenable for defining a robust coarsening procedure. On the one hand, Figure 12 shows that their is no

obvious definition for lateral fluxes at the cell boundaries if the cell resulting from the averaging of several T-cells is centered at a T-point. On the other hand, a consistent treatment of north-fold boundary condition in the HR and CRS grids requires that both grid share the same pivot point at the north-fold boundary (as described in (Madec (2008)), see in particular Fig. 8.4). This is why it is not possible to use an averaging procedure where the cell resulting from the averaging of T-cells is centered at a F-point.

The present implementation is even more limited since only a coarsening factor of 3 is allowed. This limitation is related to constraints associated with multiprocessor applications. Indeed using a larger coarsening factor would require increasing

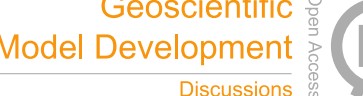

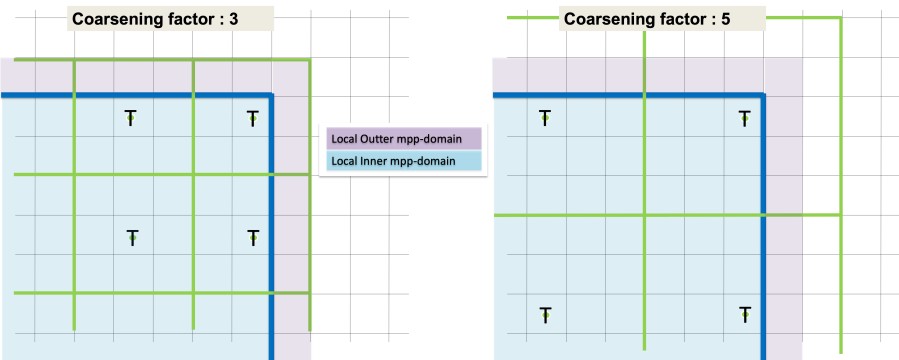

**Figure 13.** Position of coarsened cells (green) on its high-resolution grid local mpp-domain, for a 3-coarsening factor (left) and 5-coarsening factor (right).

the width of MPI overlapping areas (halo ghost cells for data exchanges with neighboring processors). This is illustrated in Figure 13, which shows that using a spatial coarsening factor of 5 would require extending the MPI overlapping area with an extra grid point in each direction.

But, as for NEMO3.6, the width of MPI overlapping areas is limited to one grid point in each direction.

A natural perspective of this work would be to add time sub-sampling. Indeed, in its the present form, our multi-grid algorithm uses the same time-stepping for both the HR and CRS grids. But the integration on the CRS grid could probably be carried out with a larger time step which would further reduce the computational cost of tracer advection. As described in (Madec (2008)) section 3.2 and (Leclair and Madec (2009)), NEMO currently uses a leap-frog time-stepping scheme with

a Robert-Asselin filter. In this context extending our algorithm to the discretization in time is non-trivial because it would require keeping in memory several consecutive occurrences of the prognostic variables and grid cells volumes. Allowing time coarsening would be greatly simplified with a 2 level time-stepping scheme in NEMO (which is planned for NEMO v5.0).

## 7   Conclusions

An algorithm based on multiple grids has been proposed and implemented in NEMO 3.6 ocean model for accelerating the

integration of ocean biogeochemical models. The core of the algorithm is to compute the evolution of biogeochemical variables on a grid which resolution is degraded by a factor 3x3 with respect to the dynamical fields. We have described in detail the operators that allow to switch from the high resolution grid (HR) to a coarser resolution grid (HR) and how to define optimally the evolution operators on the coarse grid based on information at high resolution. We have in particular described how several vertical scale factors should be introduced and described the different options for the treatment of vertical physics on the coarse

grid. A series of numerical tests performed under realistic conditions has been carried out for identifying how to optimally represent vertical mixing on the coarser grid.





The solutions computed with the proposed full algorithm have been compared to solutions obtained using a high resolution grid for both dynamics and passive tracers (HR configuration, $1/4°$ resolution), and solutions obtained using the lower resolution grid for both dynamics and passive tracers (LR configuration, $3/4°$ resolution) in a series of global ocean model experiments. We have shown that the proposed method provides solutions at global scale that are notably improved as compared to LR solutions and comparable to HR solutions. This confirms our working hypothesis that fluctuations of dynamical quantities close to the HR grid size have negligible impact on tracer transport.

The evaluation of computational performances shows that the multi-grid approach does not reduce the computing time in the case of a single passive tracer because of the overhead associated with the definition and computation of dynamical quantities on the coarse grid. However, the reduction of elapsed time can be substantial when the algorithm is applied to multiple tracers as in the case of a comprehensive biogeochemical models. The proposed algorithm allows to reduce by a factor 7 the overhead associated with running a full biogeochemical model with 24 tracer fields in NEMO simulations.

Several possible directions for improving further the performances of the algorithm have been identified but they may require important changes to NEMO code. Increasing the width of MPI overlapping areas in NEMO would allow to increase the spatial coarsening factor (now limited to 3 in the present version). In addition, the use of more selective coarsening operators (Debreu et al. (2008)) would bring the coarsened solution even more close to the uncoarsened solution. Their larger spatial stencil would however bring similar issues as for the coarsening factor limitation in a multi-grid processor environment. Extending our approach to the discretization in time is also a natural direction. Using a 2 level time-stepping scheme instead of the leap-frog time-stepping scheme currently used in NEMO would greatly simplify such a development.

*Code availability.* The code containing the multi-grid algorithm has been developped in the NEMO OGCM framework (https://www.nemo-ocean.eu/). This capacity is available on a development branch of the nemo 3.6 stable release and is available after registration at: https://forge.ipsl.jussieu.fr/nemo/browser/NEMO/branches/2018/dev_r5003_MERCATOR_CRS. The NEMO source code is freely available and distributed under CeCILL license (GNU GPL compatible). The following cpp keys have been used to compile the code :

$key\_dynspg_ts, key\_lim3, key\_vvl, key\_ldf slp, key\_traldf\_c2d, key\_dynldf\_c2d,$

$key\_zdf tke, key\_mpp\_mpi, key\_iomput, key\_nosignedzero, key\_top, key\_my\_trcandkey\_xios2.$

The key $key\_crs$ is added to this list to activate the multi-grid algorithm.

The exact version of the model used to produce the results used in this paper is archived on Zenodo at https://doi.org/10.5281/zenodo.3615356, as are input data and scripts to run the model and produce the plots for all the simulations presented in this paper (citation).

*Data availability.* The model outputs used in this paper are available at https://doi.org/10.5281/zenodo.3547421.





| HRCRS configuration diffusion coeffficient value | RMSE |
|:---:|:---:|
| 300 $m^2/s$ | 0.0823 |
| 600 $m^2/s$ | 0.0613 |
| 900 $m^2/s$ | 0.0530 |
| 1200 $m^2/s$ | 0.0535 |

**Table A1.** RMSE with HR solution, after one year of simulation.

## Appendix A: Sensitivity of tracer to diffusion coefficient

In a coupled ocean-biogeochemical model, where the ocean and biogechemical components are running at the same resolution, as the HR and LR configuration, the horizontal diffusion coeffficient $A^l$ value is the same for active tracers (temperature and salinity) and the passive tracers (the biogeochemical tracers, or the PATCH tracer here). With the multi-grid capacity, passive tracers are running at a lower resolution than active tracers. We need to evaluate the best diffusion coefficient value for the passive tracers. Here we present the results of the HR and HRCRS configurations for the PATCH experiment described in section 4.2.1.

The reference diffusion coefficient $A^l_o$ value from (9) is 300 $\mathrm{m}^2/\mathrm{s}$ for HR configuration and different simulations have been performed for HRCRS configurations with values of 300, 600, 900 and 1200 $\mathrm{m}^2/\mathrm{s}$ for $A^l_o$. The top figures in Figure A1 represent the HR solution at its resolution (left) and at the HRCRS resolution (right). The HRCRS solutions are represented above and their differences with HR in the bottom plots.

The HRCRS configurations runs with values of 300 and 600 $\mathrm{m}^2/\mathrm{s}$ for $A^l_o$ seems to be to noisy compared to the HR configuration; the maximum value of tracer inside the patch is also too high compared to HR configuration maximum. The HRCRS configuration run with a value of 1200 $\mathrm{m}^2/\mathrm{s}$ for $A^l_o$ seems to be a little bit more diffuse compared to HR configuration and the maximum value is a little bit lower. The HRCRS configuration run with a value of 900 $\mathrm{m}^2/\mathrm{s}$ gives the better reproduction to HR configuration in term of patch distribution and fine scales representation.

Table A1 gives the RMSE with HR for all experiments in the red box. The best result is obtained with a diffusion coefficient value of 900 $\mathrm{m}^2/\mathrm{s}$ and the worst with a value of 300 $\mathrm{m}^2/\mathrm{s}$. The results obtained with 1200 $\mathrm{m}^2/\mathrm{s}$ are close to the best solution.

*Author contributions.* Clément Bricaud, Christophe Calone, Gurvan Madec and Christian Ethe developed the model code. Clément Bricaud, Julien Le Sommer and Jérôme Chanut designed the experiments. Clément Bricaud performed the simulations. Clément Bricaud and Julien Le Sommer prepared the manuscript with contributions from Marina Levy, Julie Deshayes and Jérôme Chanut.

*Competing interests.* The authors declare that they have no conflict of interest.



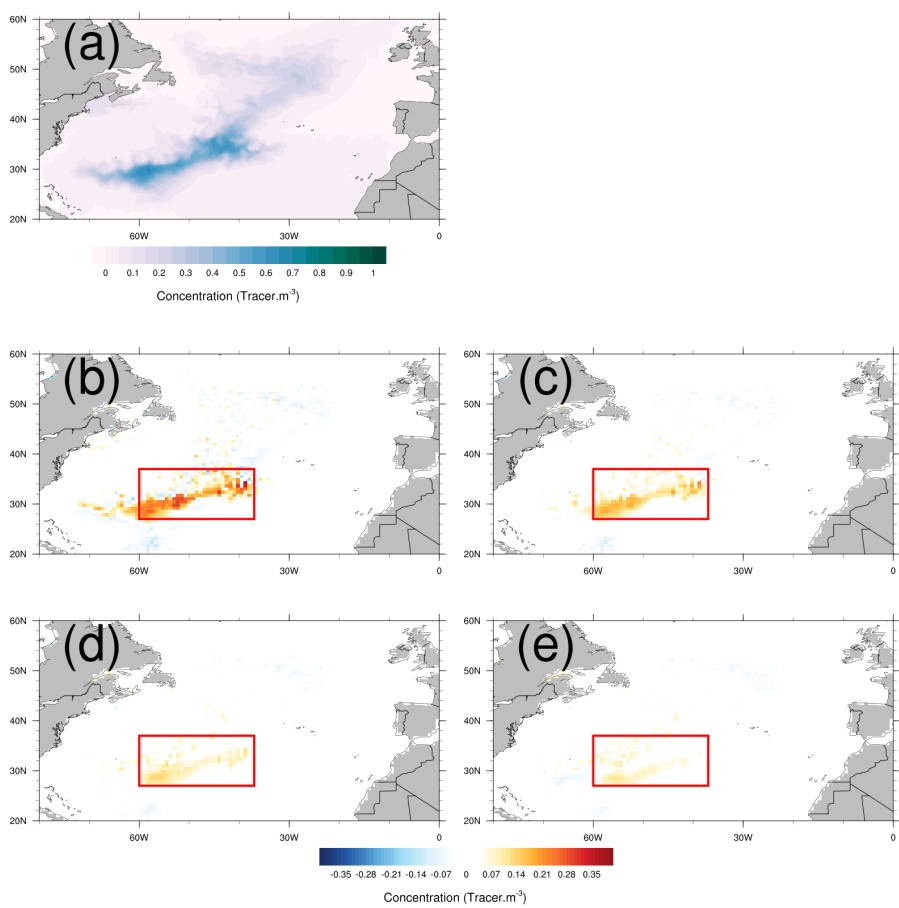

**Figure A1.** Surface values of PATCH tracer after one year for HR configuration (a); differences between HRCRS and HR configurations with diffusion=300 $\mathrm{m^2/s}$ (b) , diffusion=600 $\mathrm{m^2/s}$ (c), diffusion=900 $\mathrm{m^2/s}$ (d) and diffusion=1200 $\mathrm{m^2/s}$ (e).

*Acknowledgements.* The developper of the multi-grid algorithm for passive tracers were supported by EU MyOcean and MyOcean2 projects, the EMBRACE project (European Union's Seventh Framework Programme for research under grant agreement no 282672), then by the EU
Copernicus Marine Environment Monitoring Service (CMEMS), the French ANR SOBUMS (ANR-16-CE01-0014) and the CMEMS 22-GLO-HR project. The development and the simulation shown in this paper has been produced on the ECMWF CCA Cray supercomputer.



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
