# Peer review of "Multi-grid algorithm for passive tracer transport in NEMO ocean circulation model: a case study with NEMO OGCM (version 3.6)"

_Geoscientific Model Development, 2019_

## Referee Comment (RC1) · Anonymous Referee #1 · 12 Mar 2020

The manuscript presents a method for simultaneously applying spatial grids of multiple resolution to a popular ocean model for the purposes of reducing the computational cost of running complex biogeochemical modules. It estimates a factor 6.7 reduction in cost while achieving similar results to the single-grid high resolution version. Improving computational efficiency in models with large user groups and code bases is a priority in earth system modelling. This manuscript represents a substantial contribution to the science that is within the scope of GMD. The authors use valid methods to assess their model at a variety of spatial scales and provide sensitivity tests and examples to demonstrate their decision-making. Results are presented clearly and appropriately. Model code and other necessary scripts are provided with instructions.

[Figure]

Specific comments:

Climate is mentioned a couple of times in the manuscript, but what is demonstrated in this study is that a very slow model can be made faster (but, relative to e.g. EMICS, it is still very slow). The general motivation of making a model faster is clearly laid out in the Introduction, but even with the demonstrated improvement in runtime, as a reader I am left with uncertainty exactly what is gained with a 3X faster NEMO+PISCES. Some discussion of this in Sections 6 or 7 could help to clarify e.g. 1) does this improved runtime move NEMO+PISCES into a new category of earth system models? 2) what new kinds of applications are made possible by the improved runtime?

Simulations of 1 year are compared, but how close is the model to steady-state after only 1 year if the model is started from rest? A statement regarding why 1 year of simulation is selected for the comparison would be informative.

Some of the language needs revisiting, please see below.

Technical corrections:

P1L4: "to compute" should be "the computation of" P1L7: "allows to reduce" should be "reduces" P1L8: "tracers" P1L10, P1L11: "factor of 3";"factor of 7" P1L13: "Propositions for further reducing this cost are discussed." P2L31: run a spell-check for "operationnal" and "pannel" P2L34: "ecosystems" P2L45: "and is used in the" P2L47: This sentence is confusing. I suggest "The concept of effective resolution of physical ocean models provides a theoretical justification..." P3L50: "observation" P3L53: "experiments", "difference" P3L54: "with the velocity field...has been..." P3L65: "in particular detail" P3L69: "grid in which" P3L72: "by the hydrodynamical" P4L98: "consists of" P4L101: "that will be discussed" P6L127: "surface area" P6L135: remove "autoref" P7L137: "of the multi-grid... consists of defining..." P9L167: "very sensitive to" P9L170: "such a method" P10L181: "using the multi-grid" P10L186: "use the domain" P12L224: "Throughflow" P12L225: "close to" P13L251: remove "grid" before "multi-grid" P13L253: remove "subsection" P15: please make the x and y values larger in Fig.5 b,c,d P15: please explain in the text why HRCRS is missing from Fig.5 b,c,d at the smaller spatial scales. For example, it appears both LR and HR can simulate low KE but HRCRS cannot. P16L290: remove "subsection" P16L293: remove "appendix" P17L310: "Despite the degraded spatial resolution operating on the velocities..." P17L311: remove extra "the" P17L316: "responsible for...passive tracer..." P17L317: "...but it might be computed on..." Although, as I understand the vertical diffusion coefficient is calculated on HR and coarsened using the several methods tested. A better phrasing might be: "and coarsened to the HRCRS grid" P18L320: remove extra "section" P18L327: "results more comparable to LR" P18L329: "skill" P18L335: "extends" P18L338: "...we assess the...at simulating the...the Ross..." P19L344: "solutions...performances than the..." P19L347: "Figure 9 show...HRCRS solution..." P19L351: "...the convection..." P20L362: "caused by" P20L363: "except in the area" P22L376: "...only with large..." P23L381: "the coarsened grid" Table 4: "with/without" should be "without/with". What are the time units? P23L384 and Table 4 caption: Do you mean row? P23L386: "allows us to reduce" P23L392: remove extra "table" P23L395: "than the ocean/sea..." Table 5: "without versus with" P24L399: "their" should be "there" P24L402: "both grids share" Figure 13: "Outer" P25L411: "perspective" should be "extension"; "in its present form" P25L421: "which resolution" should be "whose resolution" P25L422: "allow the switch" P26L435: "...the reduction of elapsed time might be substantial..." (An estimate has been provided but not proven) P26L436: "case of comprehensive"; "allows us to reduce..." P26L438: "for further improving the performance" P26L439: "allow us to increase" P26L441: "even closer to" P27L458: "configurations" P27L467: "configuration"..."too noisy" P27L468:"of the tracer" P27L471: "in terms of the " P28L478: "development"

---

## Referee Comment (RC2) · Sergey Danilov (Referee) · 20 Mar 2020

It is a well written manuscript, and I am happy to recommend it for the journal. Surely, the central part of the manuscript deals with just the coarsening procedure. However, there are many accompanying issues such as vertical diffusivity, isoneutral slope, vertical cell size, etc., which have to be addressed simultaneously with coarsening; namely the description of how they are addressed is the most valuable part of the manuscript. No numerical operator is accurate at the grid scale, so that advection of numerous tracers at the original resolution in reality only kills computational resources without making tracers any more accurate compared to the case when they are advected with

reasonably coarsened velocities. One still needs extra resolution for ocean dynamics to reduce overall dissipation, but the effective resolution, as mentioned in the manuscript is much coarser than the grid scale. I think the approach proposed in the manuscript is a very good way to limit overuse of computational resources, especially given the present tendency of moving to 1/4 or 1/12 degree resolution in climate research.

Some minor points:

General: In many figures the axes or legends are hardly readable, the font size has to be increased.

line 2 running models is expensive or cost of .... is large line 7 allows to –> allows someone to (also several times in the text) line 43 remove 'also' 54 inferior to –> smaller than

Formula 2 and 3 — explain that i,j are the horizontal and k vertical indices

103 akin –> scalar

135 edit

160 the HR buoyancy?? What is coarsened Nˆ2 or isoneutral density?

187 jpk/HR

193 called to memory –> made available?

214 upfront? – just omit 215 or 225 close 236 This sentence can be omitted 249 Is e_max just equatorial resolution? 260 Why eddy? 270 adjust semicolumn 282 adjust ,as 303 plays on important role on? Just omit. ...resolution influences ... 338 remove 'propose' here and in several other places. You do not propose, you already did. 377 performance 390 as much resources as 410 one grid point — does it also mean that the third order upwind schemes are not allowed? Of course this will not be a limitation in future.

---

## Author Comment (AC1) · 15 May 2020

**Author comments**
Multi-grid algorithm for passive tracer transport in NEMO ocean circulation model:
a case study with NEMO OGCM (version 3.6)
C. Bricaud et al
submitted to Geoscientific Model Development, manuscript ID:gmd-2019-341

Corrections consecutive to referees' comments are highlighted in red.

The manuscript presents a method for simultaneously applying spatial grids of multiple resolution to a popular ocean model for the purposes of reducing the computational cost of running complex biogeochemical modules. It estimates a factor 6.7 reduction in cost while achieving similar results to the single-grid high resolution version. Improving computational efficiency in models with large user groups and code bases is a priority in earth system modelling. This manuscript represents a substantial contribution to the science that is within the scope of GMD. The authors use valid methods to assess their model at a variety of spatial scales and provide sensitivity tests and examples to demonstrate their decision-making. Results are presented clearly and appropriately.Model code and other necessary scripts are provided with instructions. Some of the language needs revisiting, please see below.

**Reply:** We thank the reviewer for the helpful feedback on our manuscript.

Specific comments: Climate is mentioned a couple of times in the manuscript, but what is demonstrated in this study is that a very slow model can be made faster (but, relative to e.g. EMICS, it is still very slow). The general motivation of making a model faster is clearly laid out in the Introduction, but even with the demonstrated improvement in run time, as a reader Iam left with uncertainty exactly what is gained with a 3X faster NEMO+PISCES. Some discussion of this in Sections 6 or 7 could help to clarify e.g. 1) does this improved run time move NEMO+PISCES into a new category of earth system models? 2) what new kinds of applications are made possible by the improved run time?

**Reply:** The complexity of the model doesn't change. By reducing the computational cost of the biogeochemical component, the algorithm will to advect passive tracers with high resolution dynamic and to integrate it over longer periods.

10 **Action:** A paragraph is added in the section 7.

Simulations of 1 year are compared, but how close is the model to steady-state after only 1 year if the model is started from rest? A statement regarding why 1 year of simulation is selected for the comparison would be informative.

**Reply:** Indeed the comparison of simulated tracer distribution are performed on tracer fields that have not reached statistical equilibrium yet, would would take centuries to millennia of model integration. But after one year of integration, the tracer distribution have been transported and stirred during several mesoscale eddy turnover time-scales so that significant differ-
15 ences can be detected across the simulations. We therefore believe that it is not necessary to wait until statistical equilibrium for comparing tracer distributions. Our approach is in line with the experimental protocols of many studies on tracer transport by mesoscale and submesoscale flows. For instance Levy et al. (2012) (doi: 10.1016/j.ocemod.2012.02.004) and Smith et al. (2016) (doi:10.1002/2015JC011089) compare tracer distributions after only several days of model integration, because this horizon is still longer than the typical timescale for evolution of the velocity field in their experiments. We have included a
20 couple of sentences in the text for explaining this choice of comparing tracer distributions after 365 days.

Technical corrections:

P1L4: "to compute" should be "the computation of"

P1L7: "allows to reduce" shouldbe "reduces"

P1L8: "tracers"

P1L10, P1L11: "factor of 3";"factor of 7"

P1L13: "Propo-sitions for further reducing this cost are discussed." P2L31: run a spell-check for "op-erationnal" and "pannel"

P2L34: "ecosystems"

P2L45: "and is used in the"

P2L47:This sentence is confusing. I suggest "The concept of effective resolution of physi-cal ocean models provides a theoretical justification..."

P3L50: "observation"

P3L53:"experiments", "difference"

P3L54: "with the velocity field...has been..."

P3L65: "inparticular detail"

P3L69: "grid in which"

P3L72: "by the hydrodynamical"

P4L98: "con-sists of"

P4L101: "that will be discussed"

P6L127: "surface area"

P6L135: remove"autoref"

P7L137: "of the multi-grid...consists of defining..."

P9L167: "very sensi-tive to"

P9L170: "such a method"

P10L181: "using the multi-grid"

P10L186: "use thedomain"

P12L224: "Throughflow"

P12L225: "close to"

P13L251: remove "grid" be-fore "multi-grid"

P13L253: remove "subsection"

P15: please make the x and y values larger in Fig.5 b,c,d.

P16L290: remove "subsection"

P16L293:remove "appendix"

7L310: "Despite the degraded spatial resolution operating onthe velocities..."

P17L311: remove extra "the"

P17L316: "responsible for...passivetracer..."

P17L317: "...but it might be computed on..." Although, as I understand thevertical diffusion coefficient is calculated on HR and coarsened using the several meth-ods tested. A better phrasing might be: "and coarsened to the HRCRS grid"

P18L320:remove extra "section"

P18L327: "results more comparable to LR"

P18L329: "skill"

P18L335: "extends"

P18L338: "...we assess the...at simulating the...the Ross..."

P19L344: "solutions...performances than the..."

P19L347: "Figure 9 show...HRCRSsolution..."

P19L351: "...the convection..."

P20L362: "caused by" P20L363: "exceptin the area"

P22L376: "...only with large..."

P23L381: "the coarsened grid" Table 4:"with/without" should be "without/with". What are the time units?

P23L384 and Table4 caption: Do you mean row?

P23L386: "allows us to reduce"

P23L392: remove ex-tra "table"
P23L395: "than the ocean/sea..." Table 5: "without versus with"
P24L399:"their" should be "there"
P24L402: "both grids share" Figure 13: "Outer"
P25L411:"perspective" should be "extension"; "in its present form"
P25L421: "which resolution"should be "whose resolution"
P25L422: "allow the switch"
P26L435: "...the reduction of elapsed time might be substantial..." (An estimate has been provided but not proven)
P26L436: "case of comprehensive"; "allows us to reduce..." P26L438: "for furtherimproving the performance"
P26L439: "allow us to increase"
P26L441: "even closerto"
P27L458: "configurations"
P27L467: "configuration"..."too noisy"
P27L468:"of thetracer"
P27L471: "in terms of the "
P28L478: "development".

**Action:** Spelling mistakes are corrected in red in the text.

**Action:** In fig05, X and Y values font is greater now.

25 **Action for"Table4: What are the time units":** The unit is added in the text caption.

**Action for "Figure 13: "Outer"":** The figure 13 is corrected

P15: please explain in the text why HRCRS is missing from Fig.5 b,c,d at the smaller spatial scales. For example, it appears both LR and HR can simulate low KE but HRCRS cannot.

**Reply:** HRCRS and also LR are missing the smaller spatial scales. Power spectra are computed along the horizontal coordinates of the model grids. For each spectrum, the minimum wavelength corresponds to twice model grid spacing: $2*\Delta x$. HRCRS
30 velocities are computed on the LR grid. HR grid resolution is around 25 km and LR/HRCRS grid resolution is around 75 km. So for HR, $1.(2*\Delta x) \simeq 2.e-2$ and for LR/HRCRS, $1.(2*\Delta x) \simeq 6.e-3$

**Action:** Explanations are added in the text.

**Referee 2**

It is a well written manuscript, and I am happy to recommend it for the journal. Surely,the central part of the manuscript deals with just the coarsening procedure. However,there are many accompanying issues such as vertical diffusivity, isoneutral slope, vertical cell size, etc., which have to be addressed simultaneously with coarsening; namely the description of how they are addressed is the most valuable part of the manuscript.No numerical operator is accurate at the grid scale, so that advection of numerous tracers at the original resolution in reality only kills computational resources without making tracers any more accurate compared to the case when they are advected with reasonably coarsened velocities. One still needs extra resolution for ocean dynamics to reduce overall dissipation, but the effective resolution, as mentioned in the manuscriptis much coarser than the grid scale. I think the approach proposed in the manuscript is a very good way to limit overuse of computational resources, especially given the present tendency of moving to 1/4 or 1/12 degree resolution in climate research.

**Reply:** We thank the reviewer for his constructive comments on the manuscript.

Some minor points: General: In many figures the axes or legends are hardly readable, the font size has tobe increased.

**Action:** Legends and/or axes font size is increased for fig05, fig08 a,d fig09.

line 2 running models is expensive or cost of .... is large
line 7 allows to –> allows someone to (also several times in the text)
line 43 remove 'also'
54 inferior to –> smaller than
Formula 2 and 3 — explain that i,j are the horizontal and k vertical indices
103 akin –> scalar
135 edit
160 the HR buoyancy?? What is coarsened $N^2$ or isoneutral density?
187 jpk/HR
193 called to memory –> made available?
214 upfront? – just omit
215 or
225 close
236 This sentence can be omitted
249 Isemax just equatorial resolution?
260 Why eddy?
270 adjust semicolumn
282 adjust,as
303 plays on important role on? Just omit. ...resolution influences ...
OK 338 remove'propose' here and in several other places. You do not propose, you already did.
377 performance
390 as much resources as

**Action:** Spelling mistakes are corrected in red in the text.

> 410 one grid point. does it also mean that the third order upwind schemes are not allowed? Of course this will not be a limitation in future.

**Reply:** The coarsening operations are computed over 3 grid points. The scheme order has no influence on coarsening operators stencil.

[revised manuscript text omitted]

---

## Author Response (AR2)

**Author comments**
Multi-grid algorithm for passive tracer transport in NEMO ocean circulation model:
a case study with NEMO OGCM (version 3.6)
C. Bricaud et al
submitted to Geoscientific Model Development, manuscript ID:gmd-2019-341

This document reports on the corrections consecutive to the Topical Editor's comments. Corrections are included in the revised file.

**Topical editor:**

5

Comments to the Author: Dear Dr. Bricaud, please improve the figure quality, especially for Fig. 5,6,7,8,9,10 and Fig A1. They have too small text (like colorbar label, axis label or imbedded text), or the plots do not look sharp even when zoomed to 150% on screen. Best wishes Qiang

**Reply:**

The size of characters for colors bar labels, legends, X/Y axis titles have been increased. Scaling of plots have been changed

10  to improve the sharpness. For some plots, the size has been increased.